# Ribosomal Dysfunction Is a Common Pathomechanism in Different Forms of Trichothiodystrophy

**DOI:** 10.3390/cells12141877

**Published:** 2023-07-17

**Authors:** Gaojie Zhu, Fatima Khalid, Danhui Zhang, Zhouli Cao, Pallab Maity, Hans A. Kestler, Donata Orioli, Karin Scharffetter-Kochanek, Sebastian Iben

**Affiliations:** 1Department of Dermatology and Allergic Diseases, Ulm University, 89081 Ulm, Germanydanhui.zhang@uni-ulm.de (D.Z.); zhouli.cao@uni-ulm.de (Z.C.);; 2Medical Systems Biology, Ulm University, 89081 Ulm, Germany; 3Istituto di Genetica Molecolare L.L. Cavalli-Sforza CNR, 27100 Pavia, Italy

**Keywords:** ribosome, trichothiodystrophy, translational infidelity, loss of proteostasis

## Abstract

Mutations in a broad variety of genes can provoke the severe childhood disorder trichothiodystrophy (TTD) that is classified as a DNA repair disease or a transcription syndrome of RNA polymerase II. In an attempt to identify the common underlying pathomechanism of TTD we performed a knockout/knockdown of the two unrelated TTD factors TTDN1 and RNF113A and investigated the consequences on ribosomal biogenesis and performance. Interestingly, interference with these TTD factors created a nearly uniform impact on RNA polymerase I transcription with downregulation of UBF, disturbed rRNA processing and reduction of the backbone of the small ribosomal subunit rRNA 18S. This was accompanied by a reduced quality of decoding in protein translation and the accumulation of misfolded and carbonylated proteins, indicating a loss of protein homeostasis (proteostasis). As the loss of proteostasis by the ribosome has been identified in the other forms of TTD, here we postulate that ribosomal dysfunction is a common underlying pathomechanism of TTD.

## 1. Introduction

Trichothiodystrophy (TTD) is a neurodevelopmental disorder with mental retardation, sulfur-deficient brittle hair, ichthyosis of the skin, recurrent infections and signs of premature aging [1]. Photosensitivity and a DNA repair defect can be detected in about 40% of TTD cases [2] that are associated with specific alterations of the basal RNA polymerase II transcription factor and DNA-repair factor TFIIH [3]. Interestingly TTD is cancer-free, in contrast to Xeroderma pigmentosum, a highly cancer-prone disease that can be provoked by mutations in the same TFIIH subunits. Therefore TTD was classified early as a transcription syndrome [4]. Half of the TTD cases are called non-photosensitive (NP-TTD) because they do not display susceptibility to sunburns by UV light and the cells do not show a defect in DNA repair of UV lesions by Nucleotide Excision Repair (NER). Nowadays several causative genes for NP-TTD are known and include the basal RNA polymerase II initiation factor TFIIEβ [5,6], four distinct tRNA-synthetases responsible for proper charging of tRNAs [7,8], the product of the C7orf11-gene TTDN1 [9] and RNF113A, a splicing factor [10]. Mutations in these broad variety of proteins with different functions provoke the same disease with sulfur-deficient hair and neurodevelopmental symptoms. Therefore, we hypothesize that there is a shared cellular pathomechanism at play. In an attempt to identify this common pathomechanism of TTD, we could already identify disturbances in ribosomal biogenesis and performance in TFIIH mutant TTD patient cells [11] and in TFIIEβ mutant patient cells [12]. TFIIH is involved in ribosomal biogenesis as an elongation factor of RNA polymerase I that transcribes the rDNA, encoding the structural and functional backbones of the ribosome, the rRNAs [13,14]. TTD mutations in TFIIH affect pre-rRNA processing and decoding accuracy of the ribosome [11]. TFIIEβ plays a critical role in distal gene occupancy of RNA polymerase I and influences the co-transcriptional assembly and maturation of the small (decoding) ribosomal subunit 40S, and mutations again affect the decoding accuracy of the ribosomes [12]. Now we intend to address whether disturbances in ribosomal biogenesis and performance are common cellular characteristics of the different forms of TTD and we focus our studies on TTDN1 and RNF113A proteins. TTDN1, also called MPLKIP, has been found in a variant form of TTD [15]; it is involved as a splicing factor in the removal of introns and its knockdown causes the preferential downregulation of long genes (preprint Townley et al., 2022). TTDN1 is associated with mitosis organization and depletion causes cytokinesis disturbances [16]. RNF113A is a component of the spliceosome [17,18], involved in the DNA repair of alkylating agents (Brickner, Soll et al., 2017) and regulates survival and differentiation of neuronal stem cells [19]. We performed a knockout of TTDN1 and a knockdown of RNF113A, and found strikingly similar cellular responses with respect to ribosomal biogenesis and performance. The impairment of both factors severely reduced the expression of UBF, a central RNA polymerase I transcription factor, it reduced RNA polymerase I transcription and impacted on ribosomal performance and proteome stability

## 2. Material and Methods

Reagents and resources: Antibodies and oligonucleotides are given in Appendix A.

### 2.1. Cell Culture

Healthy WT controls and patient cells (C3PV, 31PV, 9PV) were grown in DMEM (41965–039, Gibco, Grand Island, NE, USA) supplemented with 10% FBS and 1% penicillin-streptomycin. shRNA-transfected human NP-TTD fibroblasts (C1, sctrl) were cultured in DMEM (41965–039, Gibco) supplemented with 10% FBS and 1% penicillin-streptomycin, and 1 µ/mL puromycin. All cells were cultured at 37 °C and 5% CO_2_. Cell lines used for this study are listed in Appendix A.

### 2.2. Lentivirus Generation, Lentiviral Transfection

Lentiviral transfection was performed using bacterial glycerol stocks of three SMARTvector™ lentiviral shRNA expression plasmids and one TurboGFP control plasmid of SMARTvector™ Lentiviral Controls (Dharmacon™ Horizon Discovery Ltd., Cambridge, UK) (Table 1). This vector contains hCMV, RPRE, tGFP, IRES and puromycin sequences.

The HEK293T cells were transduced with lentivirus-mediated RNF113A shRNA or scramble-shRNA [multiplicity of infection (MOI) of 20], according to the manufacturer’s instructions, in order to generate viral particles. Before transfection, HEK293T cells were seeded on 15 cm^2^ plates until 70–80% confluency. The HEK293T cells were transfected with the psPAX2 and pMD2.G vector for the generation of viral particles along with pLKO.1-puro anti-eGFP. Subsequently, 24 h after transfection, the HEK293T cells were added with fresh DMEM complete media. After 24 h, the HEK293T cells were harvested and centrifuged. Viral-containing supernatant was collected and filtered through a 0.2 µm filter. After being supplemented with 12 µL Polybrene (10 µg/µL), the supernatant was applied onto the healthy fibroblast cells, C3PV, and incubated overnight. Then, on the second day, the media of the cells were exchanged with fresh media. The status of the cells was monitored and checked under GFP microscope. Then, the selection marker puromycin (Gibco™ Fisher Scientific, Munich, Germany) was added to the transfected cells for the selection of positive clones (1 µg/mL).

### 2.3. CRISPR CAS9 Knockout/Knockin Kit

A CRISPR CAS9 knockout/knockin kit was purchased from ORIGENE (KN204767) and the MPLKIP knockout was performed according to the manufacturer’s protocol. In total, 3 × 10^5^ adherent cells (HEK cells) were seeded 24 h prior to transfection in 10 cm dishes. The next day, three separate transfection solutions were generated in a small sterile tubes in the following combinations: scramble control + donor; gRNA1 + donor; gRNA2 + donor. An amount of 1 µg of one of the gRNA vectors (or scramble control) was added to 250 µL of Opti-MEM I (Life Technologies, GmbH, Darmstadt, Germany), and vortexed gently. Then 1 µg of the donor DNA was added into the same 250 µL of Opti-MEM, and vortexed gently. Two gRNA vectors and scramble control were in three separate tubes, so the gRNA efficiency could be tested individually. Following this, 6 µL of TurboFectin 8.0 was added to the diluted DNA and pipetted gently to mix completely. The mixture was incubated for 15 min at room temperature (Table 2).

Two weeks post transfection, cells were split 1:20. Depending on the selection marker (in this study puromycin), in a 10 cm dish the selection media were changed every 2–3 days, until single cell colonies were formed. Following this, the cells were isolated and analysis was completed by performing Western blot analysis and PCR.

### 2.4. FACs Single Cell Sorting

The knockdown efficiency of genetically engineered positive clonal population of RNF113A and TTDN1 knockdown cells were verified by flow cytometry. The transfected B9, B10, C1, gRNA1, gRNA2 and two sets of scrambled control cells (sctrl/ScrCtrl) were cultured on 75 cm^2^ cell culture flasks until 90% confluency. Cells were washed in PBS and flow cytometry analyses were performed in FACS buffer (PBS including 2% FBS). The cells were excited by laser at 488 nm to measure GFP emission at 530/30 nm using a BD FACSAria^TM^ ⅢCell Sorter (BD Life Sciences, San Jose, CA, USA). As a negative control, a non-fluorescent control cell line (C3PV) was used to set the gates for the capture of GFP fluorescent positive cells. The GFP^+^ cells were then collected for further culture. The data were analyzed with FlowJo Software v10.2.

### 2.5. RNA Isolation and cDNA Synthesis

RNA isolation was performed using an RNeasy Mini Kit (QIAGEN GmbH, Hilden, Germany) according to the manufacturer’s description and harvested RNA was measured by Nanodrop. Next, 1 ug of RNA was pre-incubated with 500 ng random primer p(dN)_6_ in nuclease free water and incubated for 5 min at 70 °C. Then, the sample was mixed with the reaction mix containing 0.5 µL dNTPs, 0.5 µL RNase ribonuclease inhibitor, 1 µL M-MLV reverse transcriptase and 5 µL 5xM-MLV reverse transcription buffer in nuclease-free water to a final volume of 25 µL, and incubated for 1 h at 37 °C for reverse transcription. The cDNA was stored at −20 °C for further use.

### 2.6. qPCR Assay Validation

The cDNA was diluted 1:50 prior to performing qPCR analysis. The quantitative PCR reaction was performed using a 7300 Real Time PCR System (Applied Biosystems^®^, Life Technologies GmbH, Darmstadt, Germany). The data were used to validate the knockdown of TTDN1 gene (gRNA1/gRNA2) and RNF113A gene (C1) and the expression of TTDN1 gene in patient cell line (31PV/9PV). Data were normalized to the level of Actin mRNA. Primers used for the analysis of qPCR are listed in Appendix A.

### 2.7. Real-Time qPCR Standard Curve Analysis

For real-time qPCR, 100 ng cDNA and FastStart Universal SYBR Green Master (denaturation at 95 °C for 10 s, annealing at 60–68 °C for 30 s, elongation at 72 °C for 30 s) were used. A standard curve for the oligonucleotide of interest with linear regression with R^2^-values > 0.8 was used for calculation of the absolute amount (ng) of the oligonucleotide of interest within 100 ng total cDNA. Primers used for the analysis of qPCR are listed in Appendix A.

### 2.8. Western Blot Analysis

Cells were grown on 75 cm^2^ culture flasks until 80% cell density, then harvested and lysed with 100 µL lysis buffer. Protein in the range 50–100 µg was loaded on 4–20% SDS-PAGE and transferred at 4 °C overnight to a nitrocellulose blotting membrane in transfer buffer (Tris-Glycine Buffer Pack: 25 mM Tris, 192 mM glycine, approximately pH8, 5% methanol, dissolved in 400 mL deionized water). Membranes were blocked at RT for 1 h with 5% BSA and 0.1% Tween 20 (diluted in TBS), washed in TBS, incubated with primary antibodies at 4 °C overnight, washed with TBS and incubated with secondary antibodies at RT for 1 h. Membranes were developed using Fusion Fx7 (Vilber-Lourmat, Eberhardzell, Germany). Images were processed and quantified by using ImageJ. Antibodies used for Western blot analysis are listed in Appendix A.

### 2.9. Immunofluorescence Staining

Cells were seeded on 4-well glass slides for 1 day. Cells at 80% confluence were washed with PBS, fixed with 4% paraformaldehyde (4 °C) for 12 min, washed with PBS, permeabilized with 0.3% Triton X-100/PBS and blocked at RT for 1 h with 5% BSA including 10% goat serum. Antibodies were diluted in Dako Antibody Diluent and cells were incubated with primary antibodies at 4 °C overnight in a moist chamber. Cells were washed with PBS, incubated with secondary antibodies at RT for 45 min in a moist chamber, washed with PBS and incubated at RT for 3–5 min with 1:5000 diluted DNA probe. After washing with PBS, cells were embedded in Dako mounting medium. Images were taken by confocal microscope (Zeiss AxioObserver Z1 confocal laser scanning microscope, 60× water objective (Zeiss, Jena, Germany)) and processed by using Leica Application Suite X (Leica, Wetzlar, Germany, 3.7.6.25997, https://www.leica-microsystems.com/products/microscope-software/, accessed on 10 August 2022). Antibodies used for the staining are listed in Appendix A.

### 2.10. Transfection and Luciferase Assay (Plasmids)

pGL3 wild-type luciferase plasmid, hypoxanthine-guanine phosphoribosyltransferase (HRPT) negative control, renilla luciferase plasmid and K529N (lysin AAA–Asn AAC) (49) mutant firefly luciferase plasmid were a kind gift from Andrei Seluanov (Vera Gorbunova) from the University of Rochester. In total, 15 × 10^4^ cells were co-transfected with 0.1 µg of renilla luciferase and 5 µg of firefly reporter plasmid (Neg or Mut) via electroporation using the Neon™ Transfection System (MPK1096, Invitrogen, Waltham, MA, USA) with following parameters: 1100 V, 20 ms, 2× pulses. Cells were plated in a white 96-well plate (5 × 10^4^ cells/well in 100 µL) overnight in Opti-MEM ((31985070, Gibco) (antibiotic free media). Luciferase activities were detected by using Dual-Glo^®^ Luciferase Assay System (E2920, Promega, Madison, WI, USA) according to the manufacturer’s protocol.

### 2.11. Luciferase Assay with mRNA Transfection

The translation fidelity assay via mRNA transfection includes the control reporter luciferase Renilla and experimental reporters Firefly (PC or MUT). Renilla and Firefly were expressed on one pCl-neo plasmid (kindly provided by Dr. Markus Schosserer). Plasmids were transcribed to mRNAs by using Ampli Cap-Max T7 High Yield Message Marker Kit according to the manufacturer’s protocol. A total of 10^5^ cells/well in 100 µL culture medium were seeded in a white 96-well plate and were grown overnight. Next, 500 ng mRNA/well in 50 µL Opti-MEM (31985070, Gibco, Grand Island, NE, USA) and 1 µL/well of Lipofectamine^®^ MessengerMAX mRNA Transfection Reagent in 50 µL Opti-MEM were incubated for 10 min at RT. The mRNA dilution and Lipofectamine dilution were mixed and incubated for further 15 min at RT. After removing the old media from cells, 100 µL of mRNA–Lipofectamine–Opti-MEM mixture were transferred to each well and cells were grown for 24 h. Luciferase activities were detected by using Dual-Glo^®^ Luciferase Assay System according to the manufacturer’s protocol.

### 2.12. Proliferation Analysis

To evaluate the growth of cell lines, 2 × 10^5^ cells were seeded in a 6-well culture dish. For each time point four dishes were detached using trypsin and counted using an automated cell counter (DeNOVIX Cell Drop FL, Wilmington, DE, USA). Cells were counted every 48 h over a period of 10 days.

### 2.13. Protein Synthesis Assay (OPP Labeling)

Protein synthesis was detected by using a protein synthesis assay kit, according to the manufacturer’s protocol. A total of 10^5^ cells/mL were grown in a white 96-well plate overnight. The following day, after being incubated with O-Propargyl-Puromycin (OPP) working solution for 1 h, the cells were fixed and stained with 5 FAM-Azide solution for 30 min. OPP analog was used for labelling translating polypeptide chains and 5-FAM-Azide for the subsequent detection of OPP-labelled proteins. After washing, fluorescence was detected by Varioskan™ LUX (Thermo Fisher Scientific, Waltham, MA, USA) using excitation 485 nm/emission 535 nm.

### 2.14. BisANS Assay

Cells were harvested in TNE buffer, sonicated (3 × 30 s) and centrifuged for 20 min at maximum speed in a table-top centrifuge. Protein concentration of the supernatant was measured using a Bradford assay. Next, 100 µg of protein was incubated in 2M urea for 2 h. BisANS dye was added (30 µM final concentration) and fluorescence was measured using an excitation wavelength of 375 nm and 500 nm emission.

### 2.15. Northern Blot Analysis

A total of 5 ug RNA was mixed with RNA loading dye (50% formamide, 7.5% formaldehyde, 1× MOPS, 0.5% ethidium bromide, DEPC H_2_O) and denatured for 15 min at 65 °C. All samples were cooled down on ice for 5 min and separated on a 0.9% agarose gel (1× MOPS buffer) for 2.5 h with 80 V (Biometra Standard Power Pack P25T, Göttingen, Germany). The gel was soaked in 20× SSC (3M NaCl, 0.3M sodium citrate-2H_2_O, adjust pH to 7.0 with HCL) and RNAs were transferred to Amersham Hybond membrane (RPN303S, GE Healthcare, Chicago, IL, USA) by a sandwich-capillary blot overnight. The membrane was twice crosslinked with 1200J UV and pre-hybridized with pre-hybridization buffer (50% formamide, 0.1% SDS, 8× Denhards solution, 5× SSC buffer, 50 mM NaP buffer pH 6.5, 0.5 mg/mL total yeast RNA) for 2 h at 65 °C. Probes were end-labeled by T4 polynucleotide kinase and 32P γATP and denatured at 95 °C for 10 min. The membrane was hybridized with the oligonucleotides using pre-hybridization buffer at 65 °C for 1 h and subsequently at 37 °C overnight. The membrane was exposed to X-ray film and quantified with ImageJ using Ratio Analysis of Multiple Precursors (RAMP) profiles (*). For rRNA processing pathway analysis we used probes binding to the region ITS1 (5′ GG GCCTCGCCCTCCGGGCTCCGTTAATGATC 3′) and ITS2 (5′ CTGCGAGGGAACCCCCAGCCGCGCA 3′).

### 2.16. Ratio Analysis of Multiple Precursors (RAMP)

The membranes of Northern blot analysis were exposed to X-ray films and quantified with ImageJ. The quantified value of each pre-rRNA was normalized to 47S pre-rRNA from each lane in order to eliminate the technical bias. Then the value of each pre-rRNA from the knockdown/knockout cell lines were again normalized to the corresponding scrambled control cell line. Finally, the ratios were transformed by log2. A RAMP profile is built by combining normalized precursor ratios in a single graph.

### 2.17. Polysomal Profiling

No less than 5 × 10^7^ cells were treated with 100 mg/mL Cycloheximide (C7698-1G, Sigma, Burlington, MA, USA) for 30 min at 37 °C. Cells were collected by scraping into 50 mL falcon tubes with PBS and 100 mg/mL Cycloheximide. Pelleted cells were mechanically disrupted with a needle (0.60 × 60 mm, 50 strikes) in 1–1.5 packed cell volumes of Dignam A containing 1 mM DTT, 1:50 cOmplete proteinase inhibitor cocktail mix (Roche, Basel, Switzerland) and 100 mg/mL Cycloheximide. Samples were incubated on ice for 1 min and centrifuged for 10 min at 1000× *g* at 4 °C. The supernatant was collected and centrifuged twice for 5 min at 12,000× *g* at 4 °C, in order to collect cytoextract. Next, 500 μg cytoextract was loaded on a 10–50% linear sucrose gradient (10% or 50% sucrose, 10 mM Tris pH 7.3, 10 mM MgCL_2_, 250 mM KCL, 25 mM EGTA, 1 mM DTT; obtained by Piston Gradient Fractionator™ (BioComp, Biocomp, Fredericton, NB, Canada)). After being centrifuged (OptimaTM MAX-XP Ultracentrifuge, Beckman Coulter, Pasadena, CA, USA) with 100,000× *g* at 4 °C for 3 h, the gradient was analyzed at OD260 nm by Piston Gradient Fractionator™ (BioComp). The baseline was defined by running a gradient without a load through the instrument.

### 2.18. Carbonylation Assay

A protein carbonylation assay kit was used to quantify carbonylation. Cells were lysed in 1× complete lysis mix. Samples were centrifuged for 20 min (13,000× *g*) and the supernatants diluted to 10 mg/mL and incubated with 2,4-Dinitrophenylhydrazine (DNPH) (1/3v 6M) for 10 min and further precipitated with (1/10v) 100% TCA. After centrifugation for 2 min at 13,000× *g* the pellets were washed with acetone followed by solubilization in 200 µL 6M guanidine. The OD was measured at 375 nm. The values were normalized according to the protein content, measured by BCA assay.

### 2.19. Statistical Analysis

All plots were created by using GraphPad Prism version 7.0.3. Each experiment was performed independently at least three times and for each individual experiment at least three technical replicates were used. Data are shown as mean ± standard deviation (SD). Statistical significance was calculated using an unpaired two-tailed Student’s *t*-test in GraphPad Prism software v6.05. Stars (*) in the figures represent ρ values (* = ρ < 0.05, ** = ρ < 0.01, *** = ρ < 0.001).

## 3. Results

### 3.1. Generation of Knockout and Knockdown Cells

TTD can be provoked by mutations in the hitherto insufficiently characterized proteins TTDN1 and RNF113A. We were starting from the hypothesis that all TTD cases are caused by a common cellular pathomechanism and asked the question if this pathomechanism might be disturbances in ribosomal biogenesis and function, as shown for TTD provoked by mutations in other unrelated proteins [11,12]. In two independent approaches, we reduced the expression of TTDN1 and RNF113A in different cell lines and investigated the consequences for cellular ribosomal biogenesis and function. Fortunately, we were able to recruit TTDN1 patient cell lines to control the results obtained by knockout of TTDN1. TTDN1 was knocked out in 293 HEK cells by a CRIPR/Cas9 approach, by using two guide RNAs (gRNA1, gRNA2) and a scrambled control (Appendix A). To confirm the successful knockout, the protein levels of TTDN1 were investigated by Western blot of protein lysates from the respective cell lines and patient-derived cells, and they displayed a severe reduction of TTDN1 protein in both cases (Figure 1A left panels, Appendix A). RNF113A was knocked down by lentiviral transfection of different GFP-shRNA constructs and a scrambled control in the fibroblast cell strain C3PV (Appendix A), and sorted by single-cell Facs analysis (Appendix A). The most efficient knockdown was achieved in the selected clone C1 (Figure 1A, right panel). The integration of the scrambled shRNA resulted in an overexpression of RNF113A (Appendix A); therefore, all experiments were also controlled in the parental cell line (Appendix A). However, the knockout/knockdown efficiencies were tested in qPCR analyses of mRNA expression and showed a highly significant reduction of the transcripts of TTDN1 and RNF113A (Figure 1B). In the next series of experiments, these cells were used to investigate the cellular localization of TTDN1 and RNF113A in confocal microscopy and to ask whether these proteins reside in the nucleolus, the locus of ribosomal biogenesis. Indeed, the co-staining of TTDN1 with fibrillarin, a nucleolus-specific methylase, clearly indicates a nucleolar localization of TTN-1 (Figure 1C). The co-staining is lost by knockout of TTDN1 or in patient-derived cells validating the knockout, the specificity of the antibody and the nucleolar staining. In contrast, despite a nuclear enrichment, RNF113A did not clearly co-localize with fibrillarin in the nucleolus. Again, the knockdown of RNF113A was accompanied by a reduced signal in immunofluorescence staining, again verifying the specificity of the antibody (Figure 1C, bottom row, Appendix A). These cells were now used for the subsequently performed experiments.

### 3.2. Reduced TTDN1 and RNF113A Impact on the RNA Polymerase I Transcription Factor UBF

In an attempt to refine the co-localization of TTDN1 with ribosomal biogenesis factors in the nucleolus, we performed co-staining of TTDN1 and RNF113A with the upstream binding factor UBF, a central transcription factor of RNA polymerase I. UBF transmits a broad variety of cellular signals to RNA polymerase I transcription [20] and serves as an essential structural component of active rDNA genes [21]. The co-staining with UBF revealed that RNF113A is not completely excluded from the nucleolus (Figure 2A and Appendix A), but these results await further in-depth analyses. Confocal microscopic images of UBF staining in the control and knockout/knockdown cells revealed that, concomitant with the reduced signal for TTDN1 or RNF113A, the UBF signal decreased (TTDN1 gRNA1) and was barely detectable in TTDN1 patient and in RNF113A mutant cells (Figure 2A). This suggests that RNA polymerase I transcription might be affected by the respective mutations. To confirm these observations, Western blots were performed and clearly revealed highly significantly reduced UBF protein abundance in TTDN1 mutant and RNF113A knockdown cells (Figure 2B). Moreover, UBF was also found to be reduced in immortalized TTD cells with TFIIH and TFIIEβ mutations and TFIIH-mutant patient cells (Figure 2B and Appendix A). To ask if this reduced abundance of UBF is due to a reduced protein stability or reflects a downregulated expression of UBF mRNA, we quantified the relative expression of UBF by qPCR after mRNA isolation from TTDN1 knockout and RNF113A knockdown cells. In fact, UBF mRNA was found to be strongly reduced in all affected cell strains (Figure 2C), suggesting that the lack of TTDN1 and RNF113A initiates signaling pathways that repress the expression of a key factor of RNA polymerase I transcription. These results indicate that TTDN1 and RNF113A impact on the regulation and activity of the first step in ribosomal biogenesis, transcription of RNA polymerase I, and provide evidence that this might be a general phenomenon in TTD.

### 3.3. rRNA Transcription and Ribosomal Subunits Are Differentially Affected in the TTDN1 Knockout and RNF113A Knockdown Cells

To investigate if the reduced levels of UBF affect activity of RNA polymerase I in our cellular models, we analyzed RNA polymerase I transcription initiation, elongation, the amount of processing intermediates and cellular content of mature rRNA by qPCR. The low UBF content might affect the early steps of RNA polymerase I transcription that we determined by the relative quantification of the parts of the 47S pre-rRNA that are firstly cleaved. These parts of the 47S-pre-rRNA were shown to be reduced in the affected cells (Figure 3A), irrespective of the nature of the impact, again indicating a common cellular consequence of different TTD-specific manipulations. Elongation activity of RNA polymerase I, as indicated by 28S/ETS and 5.8S/ITS2, was found to be reduced in TTDN1 knockout, but not in RNF113A knockdown cells (Appendix A). The primary pre-rRNA transcript is co-transcriptionally processed and assembled into pre-ribosomes [22]. Recent data suggest that the dynamics of processing of the primary transcript is disturbed in TTD [11,12]. Northern blot analysis with ITS1 and ITS2 probes (Appendix A) and relative RAMP (Ratio Analysis of Multiple Precursors) quantification (Figure 3B) revealed significant disturbances in the processing of the primary transcript in both types of mutant cells. The Northern blots evaluated the qPCR results showing reduced ITS1 amplification in TTDN1 cells and elevated ITS1 amplification in RNF113A cells (Appendix A). However, these assays cannot distinguish if the accumulation of processing intermediates is due to a loss of rRNA transcription processivity or to a reduced stability of processing products due to problems with ribosome assembly. To further analyze the subunit distribution and monosome/polysome formation, we performed polysomal profiling and quantified the areas under the peaks. The TTDN1 knockout cells display a reduced 40S abundance in this assay (Figure 3C upper graphs), in line with the reduced amounts of mature rRNAs in the qPCR (Appendix A). The RNF 113A knockdown reduced the peaks of monosomes and polysomes (Figure 3C lower graphs), again in line with a reduced abundance of mature ribosomal RNA in the qPCR (Appendix A). In both cases, the abundance of the mature 18S rRNA was found to be reduced in qPCR analysis. These results are in accordance with former publications by our group demonstrating that either mutation in TFIIEβ or TFIIH leading to TTD are followed by a reduced abundance of the small ribosomal subunit, that comprises the decoding center of the ribosome [11,12].

### 3.4. The Accuracy of the Translation Process at the Ribosome Is Decreased in TTDN1 and RNF113A Cells

The reduced abundance of the small, decoding subunit of the ribosome might be a common hallmark of TTD. If this reduction is accompanied by a qualitatively altered translation process, as already observed in other forms of TTD and in the related progeria Cockayne syndrome [11,12,23,24], it was subsequently investigated in a series of experiments. Taking advantage of an experimental system that detects the error rate of the translation process at the ribosome [25], we are using a reporter system based on firefly luciferase with a defined mutation in the active center of the enzyme (K529N). If the mutant luciferase is translated correctly by the incorporation of asparagine, no luciferase activity can be measured. Incorrect, error prone translation re-activates luciferase activity by the incorporation of the activating, near-cognate coded amino acid lysine (Figure 4A). By transfecting plasmids encoding a negative control and the mutant luciferase construct in the knockout/knockdown cells, we observed a re-activation of firefly luciferase activity as shown in Figure 4B. The transfected plasmid is transcribed by the cellular RNA polymerase II to mRNA that is then translated at the ribosome. To exclude that the detected error rate is a consequence of RNA polymerase II errors, the reporter plasmids were transcribed in vitro to mRNA and this mRNA was subsequently transfected into the cells. Here, a positive control with wildtype luciferase mRNA was used. As shown in Figure 4C, mutant mRNA is translated in an erroneous manner, re-activating luciferase activity specifically in the TTDN1 knockout and RNF113A knockdown cells, indicating an elevated translational error rate. These experiments clearly revealed that genetic manipulation of TTDN1 and interfering with the expression of RNF113A raises the error rate of the translation process at the ribosome.

### 3.5. Elevated Misfolded Proteins and a Reduced Protein Synthesis in the Affected Cells

Next, we were asking for the cellular outcomes of the hitherto described aberrations and investigated the stability of the proteome against unfolding, a parameter that distinguishes long-living from short living sea shells [26], and it was found to be reduced in Cockayne syndrome and other forms of TTD [11,23]. Cytoplasmic extracts were incubated with 2M urea, that partially unfolds the proteome, and the amount of exposed hydrophobic side chains were quantified by BisAns fluorescence. This method revealed a strong elevation of misfolded protein in the proteome of the manipulated cells (Figure 5A), suggesting that a significant proportion of the proteome might be affected by random amino acid exchanges, de-stabilizing the folding stability. Total protein synthesis was investigated by the use of a protein synthesis detection kit (Cayman). This method employs O-propargyl-puromycin (OPP) that is incorporated in the growing amino acid chain and leads to a stop of translation. The labeled products were then quantified by click-chemistry and indicate the amount of translation initiation. This process is found to be severely reduced by the manipulation of the TTDN1 gene or RNF113A expression (Figure 5B), suggesting that an error-prone protein synthesis process and the production of misfolded proteins could impact on the translation rate of novel proteins. Finally, we asked if we find, as in other forms of TTD, signs of protein oxidation in the manipulated cells. Measuring protein carbonylation—a consequence of translational errors [27]—unraveled elevated carbonylation in TTDN1 knockout/patient cells, but not in the RNF113A knockdown strain (Figure 5C). Taken together, these results demonstrate that the homeostasis of the proteins in TTD-like cells is severely affected in accordance with recent publications, demonstrating a loss of proteostasis in other forms of TTD.

## 4. Discussion

Multiple experimental approaches reveal disturbances in ribosomal biogenesis and function in different TTD-like conditions. In this study, we provide evidence by different assays that TTD-mimicking conditions lead to a loss of proteostasis by the ribosome. First we identified a loss of the central RNA polymerase I transcription factor UBF; that is, after knockout/knockdown of the TTD factors, it is barely detectable by immunofluorescence and Westerns blotting. This is not due to an elevated degradation and turnover of the protein, as complementary qPCR analysis revealed a downregulation of UBF mRNA, indicating a repression of UBF at the transcriptional level. Second, we can describe pre-rRNA processing disturbances by Northern blotting with two probes against processing intermediates, relative quantification and RAMP-analysis. Here, we observe in TTDN1 KO cells a predominant reduction of intermediates stained by the ITS1 probe, in line with the ITS1-qPCR quantification of these intermediates. Moreover, the knockdown of RNF113A provokes an accumulation of processing intermediates revealed by RAMP and qPCR analysis. Third, polysomal profiling and quantification showed reduced 40S subunit abundance in TTDN1 KO cells that is confirmed by qPCR analysis of 18S abundance in total RNA. QPCR analysis of the 5.8S and 28S rRNA abundance showed a reduction by KO of TTDN1 that is not mirrored by reduced 60S or monosomes and polysomes, as is the case for the RNF113A KD, where reduced rRNA abundance is reflected in reduced monosomes and polysomes. This difference could indicate different functional consequences for ribosomal assembly; whereas ribosomal assembly is clearly disturbed in the RNF113A KO, it appears to be normal in the TTDN1 KO, despite comparable reduced rRNA abundance in the qPCR. Fourth, transfecting plasmids and mRNA of luciferase reporter genes clearly demonstrate with these different but complementary methods that protein synthesis at the ribosome is qualitatively affected by the different TTD-mimicking conditions. Taken together, here we could identify a comparable cellular outcome of different manipulations of two genes that cause the same disease, suggesting that this could be the common underlying pathomechanism.

Is trichothiodystrophy a DNA repair or RNA polymerase II transcription syndrome? TTD, as the related progeria Cockayne syndrome (CS) that can also be provoked by TFIIH mutations, serves as a model disease to identify mechanisms of maldevelopment and degeneration. TFIIH mutant TTD patients display photosensitivity of the skin, explainable by a failure of DNA repair that does not lead to elevated mutagenesis, as TTD is cancer-free. Therefore it is not likely that the cells and organisms counteract these DNA damages to an extend that severely endangers the development and even survival of the whole organism. This hypothesis is supported by the fact that severely compromised DNA repair of UV lesions by NER can lead to high photosensitivity of the skin without further organismal impairments in the UV-sensitive syndrome (UVsS) [28,29]. CS cells, genetically related to UVsS as mutations in the same NER proteins, can provoke severe childhood degeneration and are additionally hypersensitive to oxidizing agents [30]: a feature shared with some TTD patient cells [31]. This feature might point towards a defective repair of endogenously provoked oxidative DNA lesions by NER as the driving force of neurodevelopmental impairments in TTD and neurodegeneration and premature aging in CS. However, a common DNA repair defect of endogenous oxidative DNA lesions in the different forms of TTD has not been identified yet. TTD as a syndrome of transcriptional defects of RNA polymerase II is mainly driven by seminal insights gained by the analysis of TFIIH and TFIIE mutations [6,32]. In these studies, the authors discuss that the transcription function of TFIIH is causally affected by TTD mutations. Mutations in GTF2H5, the TTDA subunit of TFIIH, can provoke TTD and this subunit is essential for the DNA repair function of TFIIH, but dispensable for the RNA polymerase II transcriptional functions [33,34]. In our recent publication we can show that TTDA mutant TTD cells display disturbances in ribosomal biogenesis, performance and cellular proteostasis, thus demonstrating a RNA polymerase I transcriptional function of TTDA [11]. How mutations in tRNA synthetases that are leading to the same TTD disease entity affecting transcription is not resolved yet. tRNA synthetases are responsible for the proper aminoacylation of tRNAs for the translation process at the ribosome and, therefore, essential for the fidelity of protein synthesis. In fact, mutation in a tRNA synthetase, as identified in the mouse “sticky” mutant, leads to misfolded proteins and neurodegeneration [35]. Moreover, interfering with the accuracy of protein synthesis by reducing the fidelity of translation at the ribosome provokes neurodegenerative symptoms in mice that resemble Alzheimer symptoms [36].

Neurodegeneration and ribosomal biogenesis and function. Neurodevelopmental symptoms and neurodegenerative symptoms are hallmarks of TTD and CS and could also be triggered by misfolded proteins. Accordingly, we could identify in TTD and CS a source of misfolded proteins: translational errors of ribosomes due to ribosomal biogenesis defects [11,12,23,24]. In this study, we can show that interference with the expression of two TTD causative genes severely impacts on ribosomal biogenesis, ribosomal performance and proteome stability. This adds to the growing list of TTD factors that are impacting on ribosomal biogenesis: the two factors TTDN1 and RNF113A. The only TTD factors that are hitherto not described to influence ribosomal performance are the tRNA synthetases [7,8], but as mutations in these factors might also impact on the accuracy of the translation process it is conceivable that they also lead to a loss of proteostasis by the production of misfolded proteins. In this study, we observe a striking uniform response of ribosomal biogenesis and performance to the impaired or reduced expression of two unrelated TTD factors. Both impacted on UBF (UBTF) expression, indicating that this might be a general mechanism to downregulate a ribosomal biogenesis and performance that affect proteostasis by the production of misfolded proteins. Downregulation of UBF could be a general feature of TTD cells. UBF is a central initiation factor of RNA polymerase I [20] and highly regulated by posttranslational modifications [37]. Moreover, depletion of UBF by RNAi leads to DNA damage and genomic instability independent of Pol I transcription indicating a role of UBF beyond ribosomal transcription [38]. Interestingly, mutations in this factor can provoke severe neurodegeneration in childhood [39,40,41] and knockout of UBF is embryonic lethal [42], whereas an induced knockout in adult mice provokes neurodegeneration [43]. Moreover, downregulation of UBF is an early event in aging-associated neurodegeneration of the human brain [44] and it is tempting to speculate that it might be a general response to the accumulation of misfolded proteins.

Is there a functional link between the spliceosome and ribosome biogenesis and function? But, how the mRNA splicing and DNA-alkylation repair factor RNF113A affects ribosomal biogenesis is mechanistically unresolved; however, the processes of DNA repair and mRNA splicing/rRNA processing are tightly interconnected as the transcription by the RNA polymerases might serve as a DNA-damage sensor [45,46]. Interestingly recent publications identified ribosomal synthesis and rRNA-processing proteins as interactors of the mRNA splicing factor XAB2 that is involved in the resolution of DNA-RNA hybrids, R-loops with the CS-factors ERCC1-XPF and XPG [47,48]. These recent studies imply that mRNA splicing, DNA repair and ribosomal biogenesis might be tightly interconnected. mRNA splicing is a newly identified function of TTDN1, that interacts with the lariat debranching enzyme DBR1, and its loss causes gene expression defects (preprint: Townley et al., 2022). Moreover, TTDN1 contains an aromatic residue rich prion-like domain that might be involved in lipid–lipid phase separation (preprint: Townley et al., 2022). Lipid–lipid phase separation organizes membraneless organelles like the nucleolus [49] and, thus, TTDN1 might be involved in the structural stability of these organelles. How it mechanistically interferes with ribosomal biogenesis and maturation awaits further analysis. Here, we describe an indirect or direct influence of RNF113A and TTDN1 on rRNA synthesis and maturation and identified qualitative disturbances of protein synthesis as a common hallmark of most, if not all, different forms of TTD.

## 5. Conclusions

TTD is a genetically heterogenous disease entity with a common, shared pathology. This fact let us hypothesize that there should be a common, shared underlying pathomechanism that determines a uniform outcome of different mutations. These different mutations could converge on one cellular signaling pathway whose failure explains the disease. Here, we provide additional evidence that disturbed ribosomal biogenesis and function is this common, shared pathomechanism of TTD. As the loss of proteostasis is a hallmark of neurodegenerative diseases [50,51], and we here show that it can be provoked by an impact of TTD-mimicking mutations on ribosomal biogenesis and function, we speculate that the neurodevelopmental phenotype of TTD might be provoked by ribosomal dysfunction. This novel hypothesis is supported by the discussed recent evidence from other and our groups.

## Figures and Tables

**Figure 1 cells-12-01877-f001:**
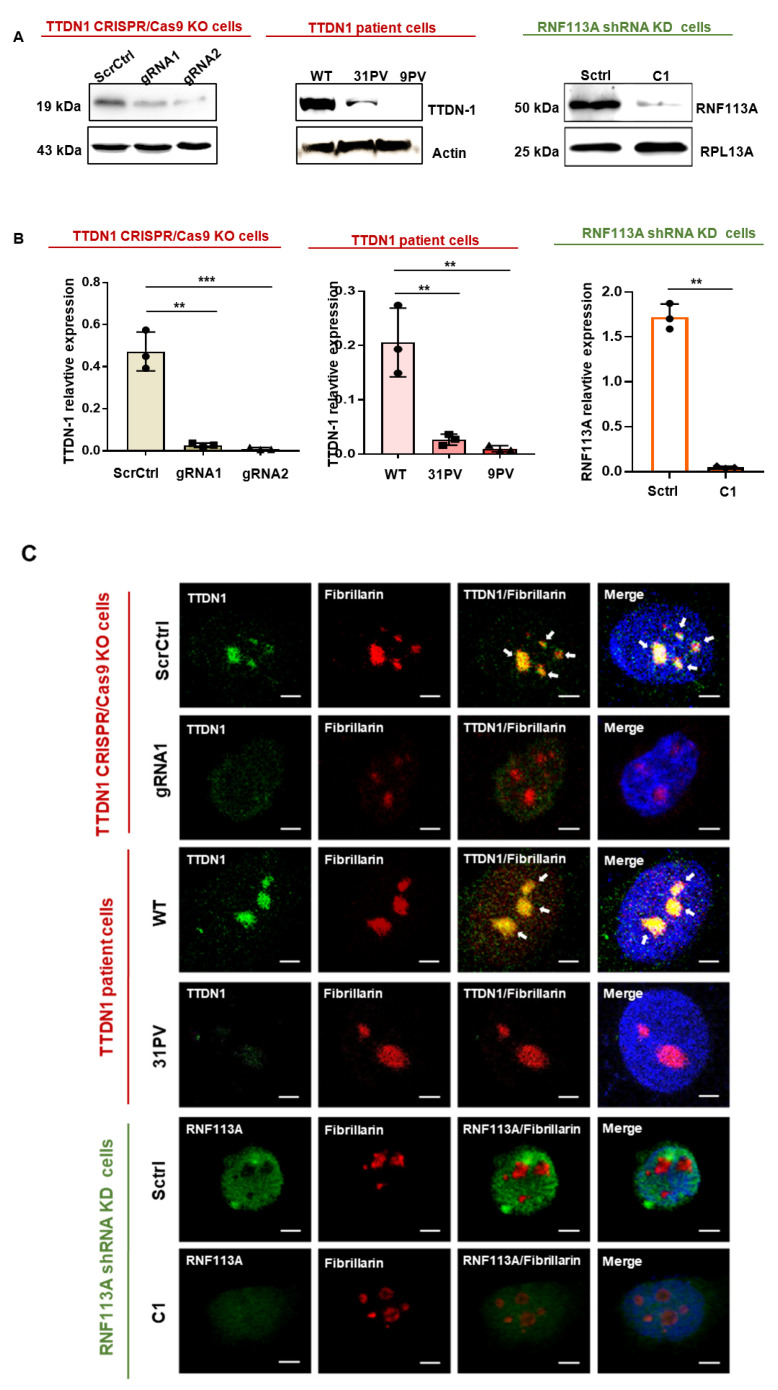
TTDN1 and RNF113A knockout/knockdown and localization. (**A**) Western blot analysis (**left**) of TTDN1 protein expression from cell lysates of CRISPR/Cas knockout and TTDN1 patient cells, actin is used as a loading control and (**right**) RNF113A protein expression in the cell lysates of shRNA knockdown cells; RPL13A is used as a control. (**B**) mRNA expression of TTDN1 gene in CRISPR Cas TTDN1 knockout and scrambled control cells, TTDN1 patient cells in comparison with wild-type cells and RNF113A relative expression in shRNA knockdown cells; actin is used as a control. (**C**) Confocal microscopy images showing co-localization of TTDN1 with nucleolus marker fibrillarin in the scrambled control and CRISPR/Cas TTDN1 knockout cells and TTDN1 patient cells in comparison with wild-type cells. Nucleolar co-localization is indicated by white arrows. Localization of RNF113A and the nucleolus marker fibrillarin in the scrambled control cells and shRNA knockdown cells. Scale bar 10 µm. Values are mean ± SD of three or more independent experiments. Asterisk in the figure represent (** *p* < 0.01, *** *p* < 0.001).

**Figure 2 cells-12-01877-f002:**
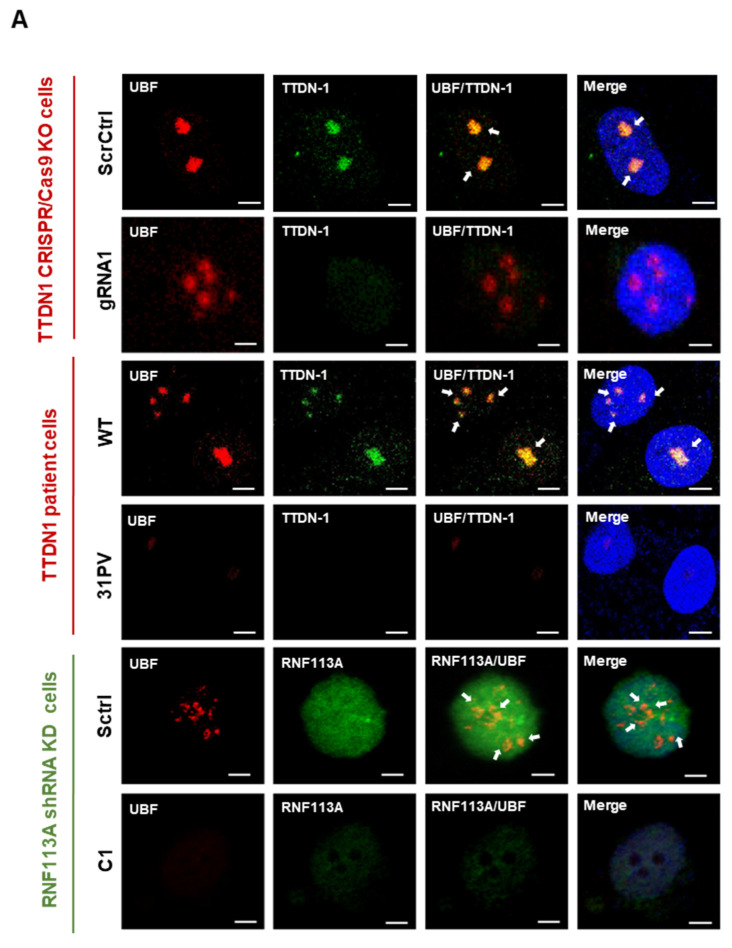
TTDN1 and RNF113A mutations lead to downregulation of UBF. (**A**) Confocal microscopy images showing localization of TTDN1 with UBF in the scrambled control and CRISPR/Cas TTDN1 knockout cells, wild-type cells and TTDN1 patient cells. Nucleolar co-localization is indicated by white arrows. Localization of RNF113A with UBF in the scrambled control in comparison with shRNA knockdown cells; scale bar 10 µm. (**B**) Western blot analysis (**left**) of UBF protein expression from cell lysates of CRISPR/Cas knockout, TTDN1 patient cells, immortalized TTD cells and primary TTD cells. Actin is used as a loading control. UBF protein expression in the cell lysates of shRNA RNF113A knockdown cells, RPL13A is used as a control. (**C**) mRNA expression of UBF in CRISPR/Cas TTDN1 knockout and scrambled control cells, wild-type cells and TTDN1 patient cells. UBF mRNA relative expression in shRNA RNF113A knockdown cells, Actin is used as a control. Values are mean ± SD of three or more independent experiments. Asterisk in the figure represent (* *p* < 0.05, ** *p* < 0.01).

**Figure 3 cells-12-01877-f003:**
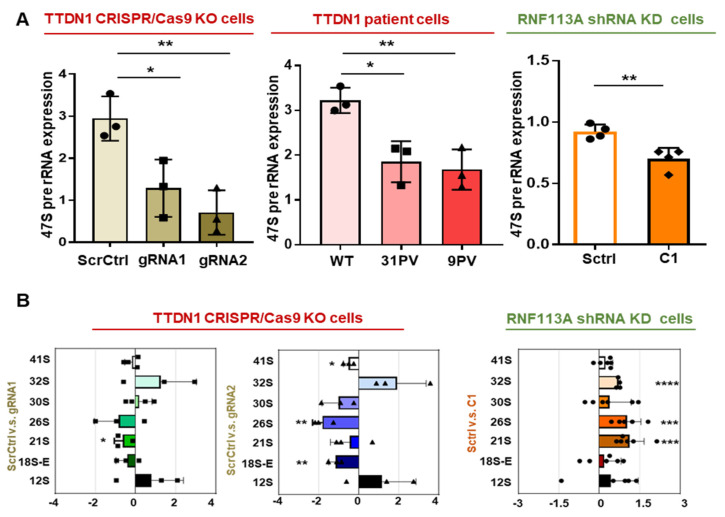
Affected ribosomal biogenesis in TTDN1 and RNF113A mutant cells. (**A**) qPCR analysis of 47S pre-rRNA expression in CRISPR/Cas TTDN1 knockout cells, TTDN1 patient cells and RNF113A shRNA knockdown cells. The values are normalized to Actin. (**B**) Northern blot of CRISPR/Cas TTDN1 knockout cells and RNF113A shRNA knockdown cells. Analysis of the Northern blots are displayed as Ratio Analysis of Multiple Precursors (RAMP) profiles [12]. Full probed membrane images are given in the Appendix A. (**C**). Polysomal analysis of CRISPR/Cas TTDN1 knockout cells, RNF113A shRNA knockdown cells and quantification of area under the peak (AUP) of 40S, 60S and 80S peaks. Values are mean ± SD of three or more independent experiments. Asterisk in the figure represent (* *p* < 0.05, ** *p* < 0.01, *** *p* < 0.001, **** *p* < 0.0001).

**Figure 4 cells-12-01877-f004:**
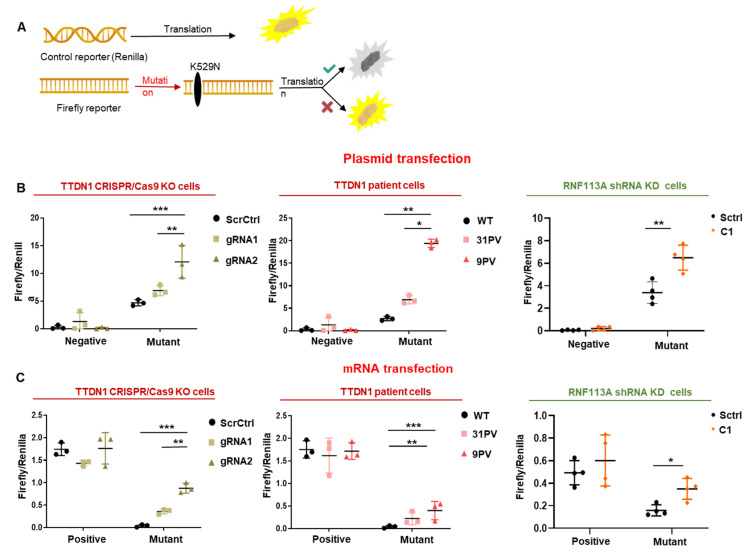
TTDN1 and RNF113A mutations disturb the quality of protein synthesis. (**A**) Schema of the luciferase-based translation fidelity assay. Cells were transfected with control and point-mutated firefly luciferase reporter. The point mutation inactivates the luciferase and by inaccurate translation the activity of the luciferase is reactivated. The values were normalized to the luminescence of renilla luciferase. (**B**) Translational fidelity was measured in scrambled control cells and CRISPR/Cas TTDN1 knockout and wild-type cells in comparison with TTDN1 patient cells. RNF113A shRNA knockdown cells were compared with scrambled control cells after transfecting cells with luciferase reporter plasmids. (**C**) Plasmids encoding for renilla and firefly luciferase were transcribed to capped mRNAs. Reporter mRNAs were transfected into CRISPR/Cas TTDN1 knockout and scrambled control cells, TTDN1 patient cells in comparison with wild-type cells and RNF113A shRNA knockdown cells in comparison with scrambled control cells. Firefly and Renilla luciferase activity was detected and relative activity calculated. Values are mean ± SD of three or more independent experiments. Asterisk in the figure represent (* *p* < 0.05, ** *p* < 0.01, *** *p* < 0.001).

**Figure 5 cells-12-01877-f005:**
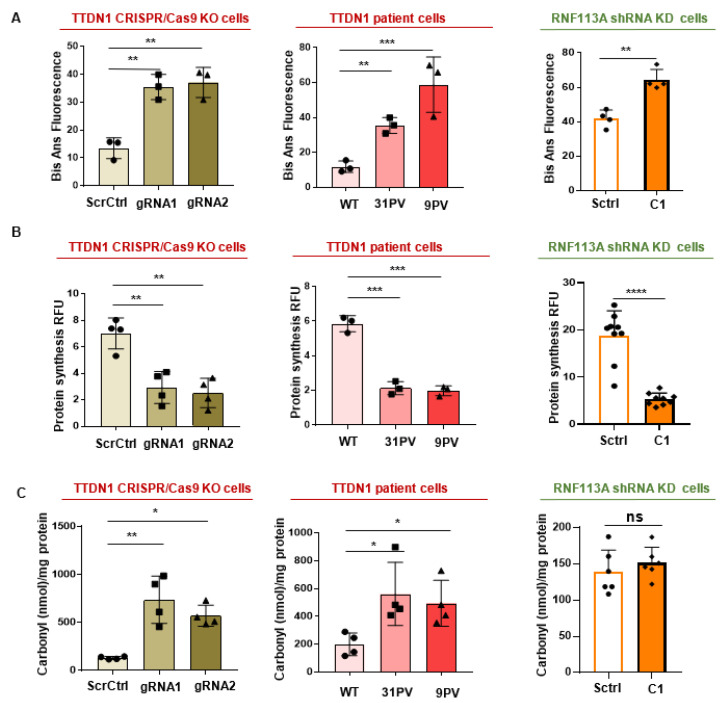
Instable proteome and reduced protein synthesis in TTDN1 and RNF113A cells. (**A**) Exposed hydrophobic residues were quantified after unfolding with 2 M urea (2 h) and labeling with BisANS fluorescent dye. Fluorescence intensity of BisANS was measured at 500 nm after excitation at 375 nm. The data show the percentage of BisANS fluorescence in CRISPR/Cas TTDN1 knockout cells and TTDN1 patient cells. Scrambled control cells served as control for RNF113A shRNA knockdown cells. (**B**) 5-FAM-Azide detection of OPP-labeled protein translation products in scrambled control and CRISPR/Cas TTDN1 knockout cells and TTDN1 patient cells. RNF113A shRNA knockdown cells were compared with scrambled control cells. (**C**) Quantification of protein oxidation, determined by the amount of carbonyl groups in CRISPR/Cas TTDN1 knockout, TTDN1 patient cells and RNF113A shRNA knockdown cells, Carbonylation was detected by the absorbance of DNPH-tagged proteins. Asterisk in the figure represent (* *p* < 0.05, ** *p* < 0.01, *** *p* < 0.001, **** *p* < 0.0001).

**Table 1 cells-12-01877-t001:** shRNA sequences of RNF113A.

	Catalog	Source Clone ID	Sequence in 5′-3′
RNF113A-shRNA1	V3SH11240-226743083	V3SVHS00_6680731	CAATGGCGTCTTCAATCCA
RNF113A-shRNA2	V3SH11240-230736642	V3SVHS00_10674292	GCGAAAGAATTGATTGCTA
RNF113A-shRNA3	V3SH11240-230369254	V3SVHS00_10306903	CCCGAGCATCTACGTGCCA
Scrambled control	SO-2952232G	SVC17010402	-

**Table 2 cells-12-01877-t002:** CRISPR CAS9 transfection mixture preparation.

Reagents	Well 1	Well 2	Well 3
Donor	1 µg	1 µg	1 µg
gRNA (1 µg each)	gRNA1	gRNA2	Scramble Control
Opti-MEM	250 µL	250 µL	250 µL
TurboFectin	6 µL	6 µL	6 µL

## Data Availability

Data are contained within the article or Appendix A.

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
