# Peer review of "Ribosomal Dysfunction Is a Common Pathomechanism in Different Forms of Trichothiodystrophy"

_cells, 2023, doi:10.3390/cells12141877_

Round 1

Reviewer 1 Report

In this article, the authors have analysed the functional consequences of mutations or reduced expression of two genes encoding TTDN1 or RNF113A, that are found mutated in some patients suffering from trichothiodystrophy (TTD). So far, TTDN1 and RNF113A were known as splicing factors. Here the authors show that TTDN1, but not RNF113A, localizes to the nucleolus. Moreover, mutations or reduced expression of TTDN1 or RNF113A genes leads to reduced UBF protein accumulation. Possibly as a consequence, mutations or reduced expression of TTDN1 or RNF113A genes leads to reduced 18S rRNA levels and alters the steady-state levels of pre-rRNA intermediates. Finally, the authors show that mutations or reduced expression of TTDN1 or RNF113A genes impairs translation efficiency and accuracy. The experiments are properly conducted and clearly presented. The results will be of interest to scientists interested in rare childhood disorders, ribosomopathies and ribosome biogenesis.

Specific points:

-          Can the reduced accumulation of proteins shown in western blots be quantified ?

-          Figure 2A: the UBF staining in CRSPR/Cas9 KO cells for TTDN1 still seems very strong, contrary to what is stated in the text. Can the authors comment?

-          The effect of RNF113A knockdown on 47S pre-rRNA levels seem rather mild (Figure 3A). Can the authors comment ?

-          The authors give the impression that 47S pre-rRNA levels are a readout for RNA pol I transcription initiation (line 326). This is misleading. Run off assays should be used instead. Please correct.

-          Overall, I feel it is a pity that the authors used qPCR rather than Northern blots to analyse pre-rRNA processing in mutant cells. Northern analysis, even if it appears as “old fashioned”, is still the best method to distinguish between the many different pre-rRNA processing intermediates, is free from amplification bias and provides information on the integrity of the different pre-rRNAs.

-          Why did the authors only analyse the levels of mature 18S rRNA (and not the levels of 28S and 5.8S rRNAs) ?

-          The fact that TTDN1, but not RNF113A, localizes to the nucleolus suggests that the effects on ribosome biogenesis in the case of TTDN1 and RNF113A are respectively direct and indirect.

Can the authors speculate as to what the function of TTDN1 in ribosome biogenesis might be ? In particular, is TTDN1 present in pre-ribosomal particles ? Any striking protein  motifs that could provide clues as to its molecular function ?

-          Minor point: “Therefore”, not “Therfor”.

Reviewer 2 Report

Trichothiodystrophy (TTD), a childhood disease with multiple symptoms, has known caused in both defective DNA repair and Pol II transcription, but other causes may also contribute. In the current manuscript, Sebastian Iben and coworkers present results suggesting that TTDN1 and RNF113A also contribute. Mutations in the former gene reduce the cell content of the UBF Pol I transcription factor and decrease rRNA production. Knockdown of RNF113A, which encodes a splicing factor, also reduces UBF content but has no strong effect on rRNA transcription and the mechanistic effects of RNA113A have not been investigated fully in this work. The experiments reported involve the analysis of CRISPR mutants and knockdown cells by confocal microscopy, qPCR, and analysis of protein synthesis, including translation accuracy, and folding. The work is structured logically but is incomplete concerning ribosome synthesis and function.

Major issues

·      Even though the qPCR results and the microscope merge results strongly indicate a defect I rRNA transcription, further analysis using northern analysis and sucrose gradient analysis combined with western blots of gradient fraction should be performed, since they have a good chance of adding important mechanistic information regarding ribosome subunit ratios, polysome formation, etc.

·      Furthermore, it is important to calculate the ratios between mature 18S and 28S to check of the subunits accumulate 1:1.

·      The microscopic images should be labeled (e.g. with arrows) for the benefit of non-aficionados

·      Decoding accuracy is affected by the 40S structure, but not necessarily by the amount of the subunit; the northern and sucrose gradient results may be useful for identifying ribosome assembly defects.

·      Break the discussion into logical paragraphs.

Minor issues

·      The manuscript has several minor language issues, e.g., “therefore” vs “therefor”, insufficient>insufficiently

·      Please explain g-samples for the non-aficionado

·      Line 290: Figure S3C>S3D).

·      Make sure that the methods are well described, including references. For example, the OPP method is not documented. Also, the length of the OPP administration may be important, since puromycin fragments are unstable. Shorter labeling times should (in theory) be more accurate.

Round 2

Reviewer 2 Report

Dr. Iben et al has submitted a revised version of Manuscript 2269489. Even though the authors have addressed by comments on the original version, the manuscript has serious issues that must be solved.

Major problems

·      Line 143: What is the justification for normalizing to actin mRNA? The abundance of actin messenger is considered a constant in normally growing cells, but is this also true in the mutated cells?

·      Figure 3: Panel C comes before panels A-B.

·      Figure 3:Panel B is truncated halfway down

·      Line 384-385: Explain “Ratio analysis” and how to interpret the graphs. It is not acceptable to ask the reader to look up an old paper in order to understand the figure

·      The Northern blot in Figure S4D is poor quality. The blots in Refs 11 and 12 are much “cleaner”. Also, the blot should be probed serially with different probes that specifically shows processing intermediates and products. This may reveal variations on the kinetics of the rRNA processing pathways.

·      Does 5.8S/ITS2 mean the whole region covered by the two elements or the ratio between the two? If the former, use 5.8S+ITS2. Same with 28S/ETS (both ETS’s?)

·      Figure 3C:

o   Where is the baseline? This must be indicated to make AUC meaningful. The instrument used for the recording of the gradient trace was clearly not adjusted to A260=0 at the bottom of the graphs

o   Adjust the y-axis to give “more height to the peaks”. The current traces are difficult to analyze.

o   RNA113A-C1: There is very limited material on the gradient. Does that mean that A260 yield in the cell lysate is very small (Methods: a constant number of cells harvested)? How do the A260 yields correlate with the growth rate? Are the effects on rRNA production an indirect effect of reduced growth rate. I.e./, is the effect on rRNA synthesis an indirect effect of reduced growth rate caused by other mechanisms than rRNA synthesis?

·      The ratios between 40S, 60S, and 80S appear to be the same in the wildtype and mutants, while the transcription of rRNA precursor is down. Does this indicate that ribosome assembly is unaffected by the mutations?

·      Discuss if you can distinguish between (i) loss of rRNA transcription processivity and (ii) reduced stability of processing products due to problems with ribosome assembly

·      Lines 421-431: Struck out. Why? Does this mean that the figure above is deleted?

Minor issues

·      Line 19:redundant with line 18

·      Lines 224-225:

o   What was the buffer used for the agarose gel?

o   The voltage setting only makes sense, if the model of the electrophoreses equipment is also given

·      Line 230: Sources of tRNA? Is it free of rRNA fragments?

·      Figure 1C and elsewhere: What red line is going halfway across the page?

·      Line 317-18: The marking of colocalized antigens by white arrows is not done consistently in all figures

·      Line 340: UBF mRNA

·      Line 366: abundance of ribosome subunits

·      Line 367: TTDN1 knockout/RNF113A knockdown cells; replace / with “versus”

·      Line 374: show>shown

Round 3

Reviewer 2 Report

Dr. Iben et al has submitted another revised version of Manuscript 2269489. Even though the authors have addressed my questions and comments, the manuscript still has serious flaws.

1.    The new northerns and the sucrose gradient baselines (Figures S4D-E) are not even mentioned in the main text. Clearly, this is not appropriate, since few readers will look for supplementary data not mentioned in the main manuscript.

2.    The definition of baselines for the sucrose gradient analysis appears to be rather arbitrary. This is a serious flaw since the AUC between the arbitrary baseline and the true baseline can seriously affect the ratio between AUC for the different ribosomal peaks, both within a gradient and between different mutants. The authors should repeat the sucrose gradient analysis and run a sucrose gradient without any load through the instrument to define a baseline. Then run the actual experimental gradients through the instrument without adjusting the baseline.

3.    The baseline adjustment should be described in the main text.

4.    The authors write in one of their answers “Unfortunately for time constrains we were not able to perform multiple experiments to clearly answer this question”. If you are not sure that the results are reproducible, the experiment should not be included in the manuscript.

5.    The ITS1 and ITS2 northerns should be quantified and compared to the results from the PCR quantifications.

6. The consistency of results obtained by different methods should be discussed.

Author Response

Comments and Suggestions for Authors

Dr. Iben et al has submitted another revised version of Manuscript 2269489. Even though the authors have addressed my questions and comments, the manuscript still has serious flaws.

  1. The new northerns and the sucrose gradient baselines (Figures S4D-E) are not even mentioned in the main text. Clearly, this is not appropriate, since few readers will look for supplementary data not mentioned in the main manuscript.

Re.: The new northerns are now cited in the text as follows: Northern blot analysis with ITS1 and ITS2 probes (supplemental Figure S4D)…….

The sucrose gradient baseline is now mentioned in the main text, material and methods: The baseline was defined by running a gradient without a load through the instrument.

  1. The definition of baselines for the sucrose gradient analysis appears to be rather arbitrary. This is a serious flaw since the AUC between the arbitrary baseline and the true baseline can seriously affect the ratio between AUC for the different ribosomal peaks, both within a gradient and between different mutants. The authors should repeat the sucrose gradient analysis and run a sucrose gradient without any load through the instrument to define a baseline. Then run the actual experimental gradients through the instrument without adjusting the baseline.

Re.: We repeated the sucrose gradient analysis according to the reviewer´s suggestion and defined the baseline to her/his advice. Representative examples are given with baseline now in the Figure 3C. We wrote in the material and methods section: The baseline was defined by running a gradient without a load through the instrument.

  1. The baseline adjustment should be described in the main text.

Re.: The sucrose gradient baseline is now mentioned in the main text, material and methods:

The baseline was defined by running a gradient without a load through the instrument.

  1. The authors write in one of their answers “Unfortunately for time constrains we were not able to perform multiple experiments to clearly answer this question”. If you are not sure that the results are reproducible, the experiment should not be included in the manuscript.

Re.: We were now able to perform multiple experiments and reached significances in Figure 3C.

  1. The ITS1 and ITS2 northerns should be quantified and compared to the results from the PCR quantifications.

Re.: The ITS1 and ITS2 northerns are quantified and depicted in Figure 3B as RAMP profiles and are now compared to the qPCR results.  The northern blots evaluated the qPCR results showing reduced ITS1 amplification in TTDN1 cells and elevated ITS1 amplification in RNF113A cells (supplemental Figure S4A, C).

  1. The consistency of results obtained by different methods should be discussed.

Re.: We now discussed this point in the result part:  The TTDN1 knockout cells display a reduced 40S abundance in this assay (Figure 3C upper graphs), in line with the reduced amounts of mature rRNAs in the qPCR (supplemental Figure S4AB). The RNF 113A knockdown reduced the peaks of monosomes and polysomes (Figure 3C lower graphs), again in line with a reduced abundance of mature ribosomal RNA in the qPCR (supplemental Figure S4C). In both cases the abundance of the mature 18S rRNA was found to be reduced in qPCR analysis. These results are in accordance

And introduced a novel paragraph in the discussion section:

Multiple experimental approaches reveal disturbances in ribosomal biogenesis and function in different TTD-like conditions. In this study we provide evidence by different assays that TTD-mimicking conditions lead to a loss of proteostasis by the ribosome. First we identified a loss of the central RNA polymerase I transcription factor UBF that is, after knockout/knockdown of the TTD factors, barely detectable by immunofluorescence and Westerns blotting. This is not due to an elevated degradation and turnover of the protein, as complementary qPCR analysis revealed a downregulation of UBF mRNA indicating a repression of UBF at the transcriptional level. Second, we can describe pre-rRNA pro-cessing disturbances by northern blotting with two probes against processing intermedi-ates, relative quantification and RAMP-analysis. Here we observe in TTDN1 KO cells a predominant reduction of intermediates stained by the ITS1 probe, in line with the ITS1-qPCR quantification of these intermediates. Moreover, the knockdown of RNF113A pro-vokes an accumulation of processing intermediates revealed by RAMP and qPCR analy-sis. Third, polysomal profiling and quantification showed reduced 40S subunit abun-dance in TTDN1 KO cells that is confirmed by qPCR analysis of 18S abundance in total RNA. QPCR analysis of the 5.8S and 28S rRNA abundance showed a reduction by KO of TTDN1 that is not mirrored by reduced 60S or monosomes and polysomes as is the case for the RNF113A KD where reduced rRNA abundance is reflected in reduced monosomes and polysomes. This difference could indicate different functional consequences for ribo-somal assembly, whereas ribosomal assembly is clearly disturbed in the RNF113A KO, it appears to be normal in the TTDN1 KO, despite comparable reduced rRNA abundance in the qPCR. Forth, transfecting plasmids and mRNA of luciferase reporter genes clearly demonstrate with these different but complementary methods that protein synthesis at the ribosome is qualitatively affected by the different TTD-mimicking conditions. Taken to-gether we here could identify a comparable cellular outcome of different manipulations of two genes that cause the same disease suggesting that this could be the common underlying pathomechanism.